# Cardiovascular diseases among people living with HIV/AIDS in Ethiopia: A scoping review

Dawit Errega[1], Marios Kantaris[2], Andualem Deresse[3], Kaleab Berhanu[4], Yohannes Gebreegziabhere[4]*, Sisay Mulugeta Alemu[5,6]

1 School of Public Health, Liverpool John Moores University, Liverpool, United Kingdom, 2 Unicaf Health Services and Social Policy Research Centre, Larnaca, Cyprus, 3 School of Public Health, College of Health and Medical Sciences, Haramaya University, Harar, Ethiopia, 4 Asrat Woldeyes Health Sciences College, Debre Berhan University, Debre Berhan, Ethiopia, 5 Division of Policy and Implementation Research for Cancer Prevention, DKFZ Hector Cancer Institute at the University Medical Centre Mannheim, Germany, German Cancer Research Centre (DKFZ), Heidelberg, Germany, 6 Department of Health Sciences, Global Health Unit, University Medical Centre Groningen, University of Groningen, Groningen, The Netherlands

* yohannes36@gmail.com

## Abstract

### Background

Human Immunodeficiency Virus (HIV)/ Acquired Immunodeficiency Syndrome (AIDS) remains a significant burden in Ethiopia. People living with HIV/AIDS (PLWHA) are at increased risk of cardiovascular diseases due to both the virus and the side effects of antiretroviral therapy. Despite a growing body of research on cardiovascular diseases in PLWHA in Ethiopia, no review has synthesised the available evidence. This scoping review aims to summarise the existing literature on cardiovascular diseases among PLWHA in Ethiopia.

### Methods

This scoping review followed the Arksey and O'Malley (2005) framework, with enhancements from Levac et al. (2010). A systematic search was conducted across six electronic databases (PubMed, EMBASE, ProQuest, Global Index Medicus, and Web of Science) in November 2023 and updated in March 2026. Studies on cardiovascular diseases among PLWHA in Ethiopia were included. Two reviewers independently screened titles, abstracts, and full texts. A data extraction template was used, and findings were synthesised narratively. The selection process was documented using a PRISMA flow diagram.

### Results

Twenty-six studies were included, with nearly two-thirds (61.5%, n = 16) focused on hypertension. The prevalence of hypertension among PLWHA in Ethiopia ranged from 11.0 to 41.3%. Factors associated with hypertension included male gender,

**Data availability statement:** All relevant data are within the manuscript and its Supporting Information files.

**Funding:** The author(s) received no specific funding for this work.

**Competing interests:** The authors have declared that no competing interests exist.

old age, rural residence, alcohol consumption, smoking, family history, low physical activity, obesity, and HAART duration/type. Other cardiovascular conditions studied included ischemic stroke, dilated cardiomyopathy, ECG abnormalities, atherosclerotic cardiovascular disease, and pericardial effusion. Most studies were cross-sectional and institution-based.

## Conclusion

This review highlights the limited yet growing evidence on cardiovascular diseases among PLWHA in Ethiopia, with a predominant focus on hypertension. Future research should use more robust study designs, such as longitudinal and interventional studies, encompass a broader range of cardiovascular diseases, and include community-based studies to better understand the prevalence and burden of cardiovascular diseases among PLWHA in Ethiopia.

## Introduction

Human Immunodeficiency Virus (HIV) and Acquired Immunodeficiency Syndrome (AIDS) continue to pose significant public health challenges in Ethiopia. Although national HIV prevalence has declined over recent years due to the expansion of Highly Active Antiretroviral Therapy (HAART) and public health interventions, approximately 1% of the population (around 1.4 million people) is still affected, with regional variation, making the country one of the most affected countries in sub-Saharan Africa [1–3]. The epidemic's impact is not limited to health; it has profound socioeconomic consequences, including increased healthcare costs, reduced labour productivity, and rising numbers of orphans and vulnerable children [4,5].

As the life expectancy of people living with HIV/AIDS (PLWHA) increases due to improved HAART coverage, the pattern of disease burden is shifting. Non-communicable diseases (NCDs), particularly cardiovascular diseases, have emerged as critical comorbidities among PLWHA globally, including in Low- and Middle-Income Countries (LMICs) like Ethiopia. PLWHA are at increased risk of cardiovascular complications due to a complex interplay of traditional risk factors, HIV-induced chronic inflammation, and side effects of long-term Antiretroviral Therapy (ART) use [6–8]. Studies have documented increased rates of atherosclerosis [9], coronary artery disease [10], myocardial infarction [7,8], ischemic stroke [11], heart failure [12], and pulmonary hypertension [13] in this population.

In Ethiopia, where the dual burden of infectious and non-communicable diseases strains the healthcare system, there is growing interest in understanding the intersection of HIV and cardiovascular health. Several primary studies have examined this relationship in the Ethiopian context. However, the existing literature remains fragmented, with wide variation in design, scope, population characteristics, and outcome measures. To our knowledge, there is no comprehensive synthesis of existing evidence that maps the extent, range, and nature of studies conducted on this topic in Ethiopia.

A scoping review is particularly well-suited to address this gap. Unlike systematic reviews that answer narrowly focused questions, scoping reviews are designed to provide an overview of the existing literature, clarify key concepts, identify research gaps, and inform future investigations or policy directions. Given the emerging and diverse nature of research on HIV-associated cardiovascular diseases in Ethiopia, a scoping review can help stakeholders, including clinicians, researchers, and policymakers, better understand the landscape and prioritise areas for intervention or further study.

Therefore, this scoping review aimed to summarise and map the available evidence on cardiovascular diseases among PLWHA in Ethiopia. Specifically, it sought to: (1) describe the type and content of existing studies on cardiovascular diseases among PLWHA in Ethiopia, (2) report the magnitude and distribution of cardiovascular diseases within this population, and (3) identify factors associated with cardiovascular comorbidities among PLWHA in Ethiopia.

## Materials and methods

This scoping review was conducted using the methodological framework proposed by Arksey and O'Malley [14], which was later refined by Levac et al. [15] and aligned with the Joanna Briggs Institute (JBI) guidelines for scoping reviews [16]. To ensure clarity and transparency in reporting, we adhered to the PRISMA extension for Scoping Reviews (PRISMA-ScR) checklist [17].

### Identification of the research questions

The review was guided by the following research questions: (1) What are the types of studies conducted on PLWHA concerning cardiovascular diseases in Ethiopia? (2) What is the prevalence/incidence of cardiovascular diseases in PLWHA in Ethiopia? And (3) What are the key factors associated with cardiovascular diseases among this population?

Studies were included if they investigated cardiovascular disease among PLWHA, were conducted in Ethiopia, and were published in English. There were no restrictions on the study design, participant age, or year of publication. We excluded studies that focused only on risk factors (such as obesity, dyslipidaemia, or metabolic syndrome) without measuring cardiovascular disease outcomes, studies only on the health service delivery process, studies involving Ethiopian migrants living outside of Ethiopia, multi-country studies where Ethiopia-specific data could not be extracted, and commentaries, protocols, editorials, letters, conference abstracts, and anonymous reports. Studies without accessible full texts were also excluded after attempts to contact the authors.

### Identification of relevant studies

We systematically searched five electronic databases, i.e., PubMed, Embase, ProQuest, Global Index Medicus, and Web of Science, from inception to 13th November 2023, and later updated on 13th March 2026. A comprehensive search strategy was initially developed for PubMed using a combination of Medical Subject Headings (MeSH) and free text for four big terms, i.e., 'cardiovascular disease,' 'HIV,' AIDS,' and 'Ethiopia'. This strategy was then adapted for the remaining databases. In addition to electronic databases, we hand-searched the bibliographies of included studies and relevant grey literature sources, including the Addis Ababa University digital thesis repository. Full details of the search strategy are provided in S1 File.

### Study selection and appraisal

All identified records were imported into EndNote citation manager for de-duplication, and the screening process was managed using Covidence, a web-based platform for systematic reviews [18]. Title and abstracts of each study were screened independently by two reviewers (DE and YG/SMA), and any discrepancies were resolved through discussion with a third reviewer (AD). After removing irrelevant studies, full texts were systematically reviewed for further eligibility analysis. Full texts of potentially relevant articles were then retrieved and assessed for eligibility by one reviewer (DE/

SMA), with input from a second reviewer as needed. The screening and selection process is presented in a PRISMA flow diagram (Fig 1).

## Data charting process

For all included studies, we developed a standardized data extraction form to collect relevant information, including title, year of publication, aim or research questions, sampling technique, country, outcome variables, study design, study period, study participants/population, sample size, data collection method, main data analysis technique, the cardiovascular disease of interest, and main findings and conclusions. Data extraction was performed by one reviewer (DE/SMA) and checked for accuracy and completeness by another reviewer (YG/AD/KB).

## Collating, summarising, and reporting of the results

A narrative synthesis approach was used to summarise the findings due to substantial heterogeneity in study designs (e.g., cohort, cross-sectional, case report), population (e.g., patients on HAART versus not on HAART), outcomes (e.g., hypertension, ischemic stroke, atherosclerotic, and ECG abnormalities), and measurement and analysis methods. As such, meta-analysis was not feasible. The findings are presented in both narrative form and summary tables. The Preferred Reporting Items for Systematic Reviews and Meta-Analyses (PRISMA) guideline for scoping reviews [19] was strictly followed in reporting the review results (S2 File).

# Results

## Characteristics of included studies

A total of 678 studies were returned by the search, and 19 studies that met the inclusion criteria were included in the narrative synthesis; an additional 7 studies were included after the updated search (Fig 1). The list of excluded articles with the reason for exclusion is in S3 File. The total sample size represented in this review comprises 9,106 participants. Although not all studies reported participants' age, gender, and residential characteristics, available data indicate that approximately 62% were female and 74.5% resided in urban areas. Nearly all studies focused on adult populations, except for two: one was a case report involving an infant [20], and the other included only foetuses [21].

Over three-quarters (76.9%, n = 20) of the studies were conducted in only three of the nine regions of Ethiopia (i.e., Addis Ababa [20,22 27], Amhara [28–33], and Southern Nations Nationalities and Peoples (SNNP) regions) [34–41]. The average sample size across the included studies was 350 participants; 21 (80.8%) were cross-sectional, 4 (15.4%) were cohort studies, and only one study was a case report. In addition, all the included studies were institution-based and were conducted after 2011. Nearly two-thirds (61.5%, n = 16) of the included studies were about hypertension [24,28–34,36–38,40,42,43]. Four (15.4%) studies were about more than one cardiovascular disease [21,23,35,44], while the remaining studies were about atherosclerotic cardiovascular disease [22,26], EEG abnormalities [39,41], pulmonary hypertension [25], and ischemic stroke [20] (Table 1).

## Prevalence of cardiovascular disease among PLWHA

Sixteen studies of the included studies were about hypertension [24,28–34,36–38,40,42,43], of which two reported the incidence [32,36], while the rest reported the prevalence of hypertension among PLWHA. The definition of hypertension varied across the included studies. Five studies defined hypertension as blood pressure ≥140/90 mmHg; six studies used the same threshold or included individuals taking antihypertensive medication; and three studies applied a lower threshold, defining hypertension as ≥130/80 mmHg.

The prevalence of hypertension in these studies ranges from 11.0% [37] to 41.3% [28], while the two studies reported nearly similar incidence rates, 12.0% and 13.2% [32,36]. A comparative study found a one-year incidence of uncontrolled

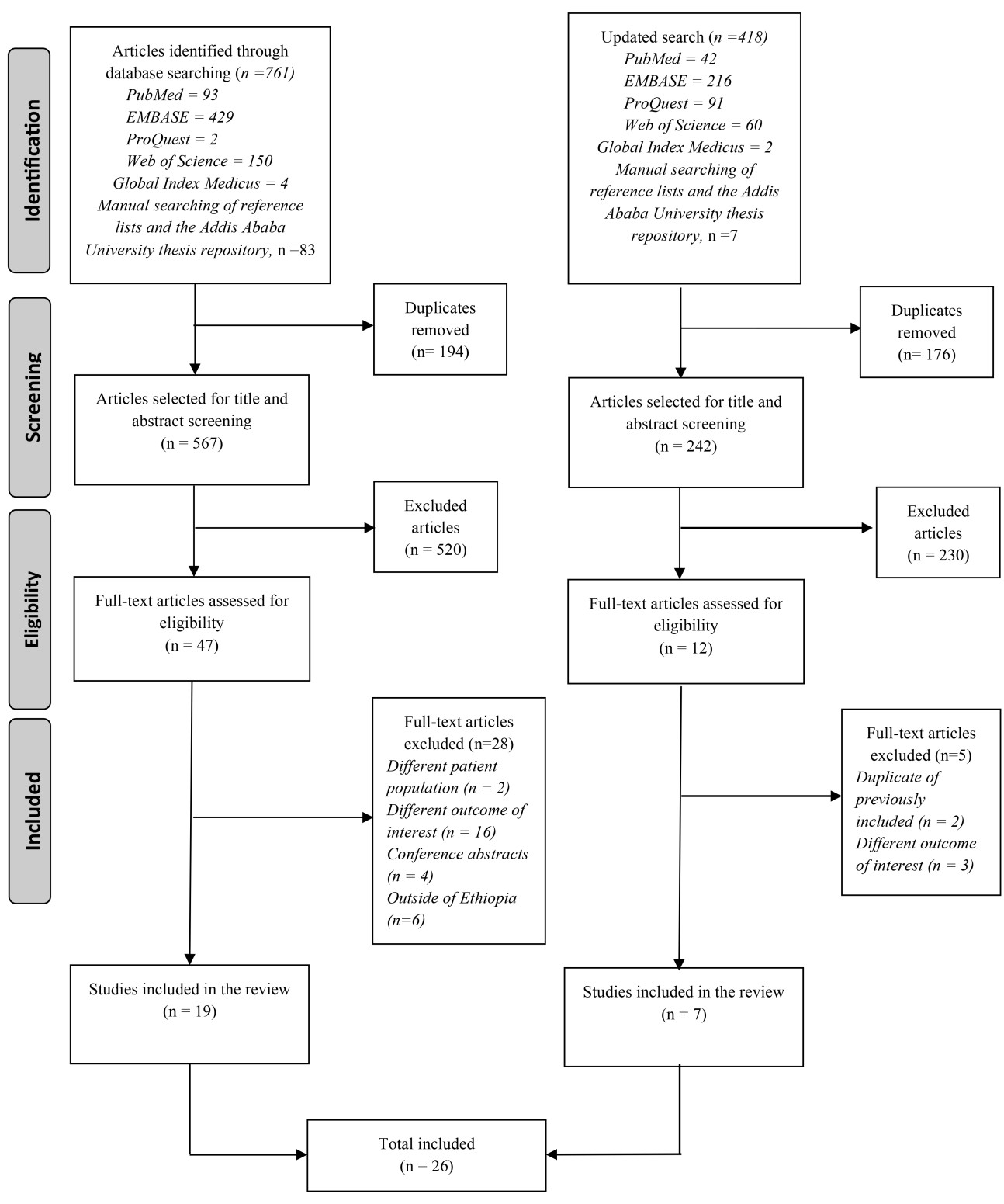

**Fig 1. Preferred Reporting Items for Systematic Reviews and Meta-Analyses (PRISMA) flow diagram of article screening and selection process.**

**Table 1. Description of the characteristics of included studies.**

| Citation | Population | Included participants | Study period | Study design | Study type | Study area | Data collection method |
|---|---|---|---|---|---|---|---|
| (Zewudie et al., 2022) [34] | Adult PLWHA on HAART | 388 | May 1 to July 1 2021 | Cross-sectional | Institution based | SNNP (Butajira General Hospital) | Structured questionnaire |
| (Woldu et al., 2021) [22] | Adult PLWHA | 314 | January 2019 to February 2020 | Cross-sectional | Institution based | Addis Ababa (Zewditu Memorial Hospital) | Structured questionnaire Document review Laboratory tests Clinical examination |
| (Woldeyes et al., 2022a) [23] | Adult PLWHA | 285 | January 6, to February 15, 2020 | Cross-sectional | Institution based | Addis Ababa (St. Paul. Hospital Millennium Medical College) | Structured questionnaire |
| Woldeyes et al., 2022b) [24] | Adult PLWHA on HAART | 333 | January 6, to February 15, 2020 | Cross-sectional | Institution based | Addis Ababa (St. Paul. Hospital Millennium Medical College) | Structured questionnaire |
| (Woldesemayat, 2020) [35] | Adult PLWHA on HAART | 382 | April 2016 | Cross-sectional | Institution based | SNNP (Hawassa University Comprehensive Special-ized Hospital) | Structured questionnaire Document review |
| (Tegene et al., 2023) [36] | Adult PLWHA on HAART | 520 at baseline | May 2019 to June 2021 | Cohort study | Institution based | SNNP (Hawassa University Comprehensive Special-ized Hospital) | Structured questionnaire |
| (Tegene and Belachew, 2016) [44] | Adult PLWHA on HAART | 280 | 16th Sept 2012–15th July 2013 | Cross-sectional | Institution based | Oromia (Jimma University Medical Centre) | Structured questionnaire ECG |
| (Sewale et al., 2020) [28] | Adult PLWHA on HAART | 412 | February to March 2020 | Cross-sectional | Institution based | Amhara (Debre Markos Referral Hospital) | Structured questionnaire |
| Mulugeta 2021 et al., 2021) [32] | Adult PLWHA on HAART | 302 | October to December 2019 | Cohort study | Institution based | Amhara (East Gojjam) | Standard checklist Blood pressure measurement |
| (Melaku 2020 et al., 2020) [45] | Hypertensive adult PLWHIV on HAART | 103 | December 2018 to February 2019 | Cohort study | Institution based | Oromia (Jimma University Medical Centre) | Structured questionnaire Document review |
| | HIV-negative hypertensive adults | 200 | | | | | |
| (Lukas 2021 et al., 2021) [37] | Adult PLWHA on HAART | 382 | January 20 to March 20, 2020 | Cross-sectional (mixed method) | Institution based | SNNP (Wachemo University Nigist Elleni Moham-med Memorial Referral Hospital) | Structured questionnaire In-depth interview |
| (Huluka et al., 2020) [25] | Adult PLWHA on HAART | 315 | June 2018 to February 2019 | Cross-sectional | Institution based | Addis Ababa (Tikur Anbessa Specialized Hospital) | Structured questionnaire ECG |
| (Getahun et al., 2020) [29] | Adult PLWHA on HAART | 560 | March 1–31, 2018 | Cross-sectional | Institution based | Amhara (Bahir Dar city) | Structured questionnaire |
| (Gebrie, 2020) [30] | Adult PLWHA on HAART | 407 | November 2018 to May 2019 | Cross-sectional | Institution based | Amhara (Debre Markos and Felege Hiwot Referral Hospitals) | Structured questionnaire |
| (GarcÃa-Otero et al., 2022) [21] | HIV exposed infants | 29 | March 2017 and November 2018 | Case-control | Institution based | Oromia (two health centres and one hospital in the city of Adama) | Structured questionnaire |
| | HIV unexposed infants | 67 | | | | | |

*(Continued)*

**Table 1.** (Continued)

| Citation | Population | Included participants | Study period | Study design | Study type | Study area | Data collection method |
|---|---|---|---|---|---|---|---|
| (Fiseha 2019 et al., 2019) [31] | Adult PLWHA on HAART | 408 | January to May 2018 | Cross-sectional | Institution based | Amhara (Dessie Referral Hospital) | Structured questionnaire |
| (Badacho and Mahomed, 2023) [38] | Adult PLWHA on HAART | 520 | January and June 2022 | Cross-sectional | Institution based | SNNP (five selected PHCs in Wolaita zone) | Structured questionnaire |
| (Ataro et al., 2018) [42] | Adult PLWHA on HAART | 425 | February and April 2017 | Cross-sectional | Institution based | Harari (Jugol Hospital) | Structured questionnaire |
| (Alemayehu et al., 2019) [20] | A 19-month-old boy | 1 | May 2018 | Case report | Institution based | Addis Ababa (Tikur Anbessa Specialized Hospital) | In-depth interview Document review Physical examination |
| (Befkadu et al., 2018) [39] | Adult PLWHA and on follow-up | 298 | November 4 to December 25, 2022 | Cross-sectional | Institution based | SNNP (Mettu Karl Special-ized Hospital) | Structured questionnaire ECG |
| (Arega et al., 2025) [26] | PLWHA on combined 1st & 2nd line HAART | 331 | September 1–30, 2024 | Cross-sectional | Institution based | Addis Ababa (Government hospitals) | Document review |
| (Jemal et al., 2025) [33] | Adult PLWHA on dolutegravir | 415 | February 5 to April 5, 2023 | Cross-sectional | Institution based | Amhara (Dessie Com-prehensive Specialized Hospital) | Structured questionnaire patient chart review |
| (Hirigo et al., 2024) [40] | Adult PLWHA taking dolutegravir | 444 | January 2023 to May 2024 | Cross-sectional | Institution based | SNNP (Six health facilities in Hawassa City) | Structured questionnaire Document review |
| (Haile et al., 2024) [27] | Adult PLWHA on HAART | 411 | December 15, 2021, to January 20, 2022 | Cross-sectional | Institution based | Addis Ababa (Alert and St. Peter's Specialized Hospitals) | Structured questionnaire |
| (Dechasa et al., 2024) [43] | Adult PLWHA on HAART | 382 | March 20 to April 14 2023 | Cross-sectional | Institution based | Harari (Public hospitals in Harar City) | Structured ques-tionnaire Document review |
| (Befkadu et al., 2024) [41] | Adult PLWHA | 96 | January 11 to March 10, 2022 | Cross-sectional | Institution based | SNNP (Mettu Karl Special-ized Hospital) | Structured questionnaire EEG |
| | HIV-negative controls | 96 | | | | | |

ECG: Electrocardiogram; HAART: Highly Active Antiretroviral Therapy; PLWHA: People Living with HIV/AIDS; SNNP: Southern Nations Nationalities and Peoples.

blood pressure in 60.2% of PLWHA with hypertension compared with 53% of people who are HIV-negative and have hypertension [45]. On the other hand, pulmonary hypertension was reported in 14% of PLWHA [25] and atherosclerotic cardiovascular disease among 11.5% to 28.7% of PLWHA [22,26].

Some studies also reported the prevalence of multiple cardiovascular disorders. For example, one study reported the prevalence of multiple echocardiographic abnormalities, such as diastolic dysfunction (30%), ischemic heart disease (19.3%), left ventricular hypertrophy (10.2%), enlarged left atrium (8.1%), pulmonary hypertension (3.6%), and pericar-dial effusion (2.1%) [23]. Another study found different cardiac abnormalities among 42.1% of PLWHA, the most common forms being diastolic dysfunction and left ventricular hypertrophy (LVH), each accounting for 8.9%, followed by systolic dysfunction (6.1%), dilated cardiomyopathy (5.4%), coronary artery disease (3.9%), pericardial effusion (2.5%), and pul-monary hypertension (1.4%) [44]. Another study on multiple cardiovascular disorders found 4.5% of PLWHA to have either hypertension, congestive heart failure, or rheumatic heart disease [35] (Table 2).

**Table 2. Main findings as extracted from the included studies.**

| Citation | Population | Disease of interest | Main Findings |
|---|---|---|---|
| (Zewudie et al., 2022) [34] | Adult PLWHA on HAART | Hypertension | - Prevalence of undiagnosed hypertension = 18.8% (95% CI: 14.7%, 23.2%)<br>- Diabetes mellitus, alcohol drinking, duration on ART, and age were significant variables for undiagnosed hypertension at p-value<0.05. |
| (Woldu et al., 2021) [22] | Adult PLWHA | Atherosclerotic Cardiovascular Disease | - Prevalence among age 20–49 (both tools) = 11.5%<br>- Prevalence among age 40–79 years (Pooled Cohort Equation) = 28%<br>- Prevalence among age 40–79 years (Framingham Risk score) = 17.7%<br>- Using both tools, advanced age, male gender, smoking, and increased systolic blood pressure were associated with an increased risk of Atherosclerotic Cardiovascular Disease |
| (Woldeyes et al., 2022a) [23] | Adult PLWHA | ECG abnormalities | - Diastolic dysfunction was common (30%), followed by ischemic heart disease (19.3%), left ventricular hypertrophy (10.2%), enlarged left atrium (8.1%), pulmonary hypertension (3.6%), and pericardial effusion (2.1%)<br>- Male gender, age > 50 years, elevated blood pressure, and elevated FBG were associated with echocardiographic abnormalities. |
| Woldeyes et al., 2022b) [24] | Adult PLWHA on HAART | Hypertension | - Prevalence of hypertension = 23.8%; metabolic syndrome = 22.8%, & Obesity = 11.1%<br>- The Framingham risk score was low in 95.9% of cases.<br>- Male gender, increasing age, high body mass index, and previous ART regimen being tenofovir disoproxil fumarate, lamivudine, and nevirapine increased the cardiovascular disease risk factor |
| (Woldesemayat, 2020) [35] | Adult PLWHA on HAART | Cardiovascular system (hypertension, CHF, RHD) | - Prevalence of cardiovascular problems = 4.5%; Hypertension = 4.2%<br>- Age, ART duration, and CD4 count were determinant factors of having any one of the non-communicable chronic diseases, while being on ART for at least for six years predicted having multimorbidity. |
| (Tegene et al., 2023) [36] | Adult PLWHA on HAART | Hypertension | - Incidence of hypertension = 12%<br>- Age, nutritional status, and regular physical exercise were reported as predictors of chronic comorbidity |
| (Tegene and Belachew, 2016) [44] | Adult PLWHA on HAART | Cardiac disorder | - Prevalence of cardiac abnormality = 42.10%<br>- The most common abnormalities were Left Ventricular Hypertrophy (8.9%) and Ventricular diastolic dysfunction (8.9%), followed by systolic dysfunction (6.1%).<br>- Dilated cardiomyopathy was significantly associated with lower CD4 counts (p<0.002), not starting on ART (p<0.05), and lower Body mass index (BMI) (p<0.05). Pericardial effusion was found among 2.5% of the individuals and was significantly associated with ART status (p<0.05) |
| (Sewale et al., 2020) [28] | Adult PLWHA on HAART | Hypertension | - Prevalence of hypertension = 41.30%<br>Age group (35, 45) and (>45) years, not having regular exercise, BMI >=25, and patients taking (2 h (TDF+3TC+ATV/r)/ 2f(AZT+3TC+ATV/r)/ 2e (AZT+3TC+LPV/ r)/ ABC+3TC+ATV/r) increased the risk of hypertension |
| Mulugeta 2021 et al., 2021) [32] | Adult PLWHA on HAART | Hypertension | - Incidence of hypertension = 13.25%, incidence rate of 16.35 per 1000 person-month<br>- Male sex, old age, WHO staging stage 3 & 4, diabetes mellitus comorbidity, ART regimen (AZT containing), and BMI (overweight and obesity) were significant predictors for the development of hypertension |
| (Melaku 2020 et al., 2020) [45] | Hypertensive adult PLWHA on HAART HIV-negative hypertensive adults | Hypertension | − 60.2% of HIV-positive and 53% of HIV-negative patients showed uncontrolled blood pressure<br>- An increased waist circumference, chronic disease comorbidity, alcohol use history, HIV infection, infrequent use of fruits & vegetables, infrequent engagement in physical exercise, and frequent use of high-fat food were independent predictors of uncontrolled blood pressure. |
| (Lukas 2021 et al., 2021) [37] | Adult PLWHA on HAART | Hypertension | - Prevalence of hypertension = 11.0% 95% CI 7.93, 14.04<br>Staying on HAART for a long time, alcohol, and the ART regimen were significantly associated with hypertension |
| (Huluka et al., 2020) [25] | Adult PLWHA on HAART | Pulmonary hypertension | - Prevalence = 14.0%<br>- None of the previously known factors, such as age, gender, CD4 level, and viral load, that are associated with increased risk of pulmonary hypertension, were related to the presence of pulmonary hypertension in this study |
| (Getahun et al., 2020) [29] | Adult PLWHA on HAART | Hypertension | - Prevalence = 14.1%<br>- Old age, BMI of 25 or more, and taking ART regimens containing TDF were factors significantly associated with HIV-hypertension comorbidity. |

*(Continued)*

**Table 2.** (Continued)

| Citation | Population | Disease of interest | Main Findings |
|---|---|---|---|
| (Gebrie, 2020) [30] | Adult PLWHA on HAART | Hypertension | - Prevalence = 14%<br>- Elementary educational status, moderate monthly income, waist circumference, taking concomitant other drug therapy, and duration of antiretroviral therapy were significantly associated with hypertension |
| (GarcÃa-Otero et al., 2022) [21] | HIV exposed infants / HIV unexposed infants | General (Cardiovascular effects) | - No significant differences were observed in the cardiovascular measurements between the groups regarding cardiac morphometry function.<br>- Regarding vascular assessment, similar results in cIMT were observed between the groups |
| (Fiseha 2019 et al., 2019) [31] | Adult PLWHA on HAART | Hypertension | - Prevalence = 29.7% (95% CI, 25.3%, 35.0%)<br>- Older age, male gender, longer duration on ART, high body mass index, and diabetes were independent risk factors of hypertension |
| (Badacho and Mahomed, 2023) [38] | Adult PLWHA on HAART | Hypertension and diabetes | - Prevalence = 18.5% (95% CI: 15.2%, 21.7%)<br>- Age >= 45 years, alcohol consumption, insufficient physical activity, BMI > 25, family history of hypertension, and diabetes were significantly associated with hypertension. |
| (Ataro et al., 2018) [42] | Adult PLWHA on HAART | Hypertension | - Prevalence = 12.7%<br>- Raised waist-to-hip ratio, raised blood glucose, increased total cholesterol, high current BMI (overweight/obesity), drinking alcohol, current CD4 count <500 cells/mL, and longer duration of HAART were significantly associated with hypertension |
| (Alemayehu et al., 2019) [20] | A 19-month-old Ethiopian boy | Ischemic stroke | - The study described the case of a 19-month-old boy who presented with a left-sided body weakness of sudden onset and concluded that extrapulmonary tuberculosis should be considered a cause of sudden focal neurologic deficits in children with endemic HIV. |
| (Befkadu et al., 2018) [39] | Adult PLWHA and on follow-up | ECG abnormalities | - Heart rate corrected QT interval prolongation was prevalent in 12.4% of participants.<br>- Being on protease inhibitors containing HAART regimen, having recent CD4 count of < 350 cells/mm3, and recent viral load of ≥ 1000coppies/ml were significantly associated with QT interval prolongation. |
| (Arega et al., 2025) [26] | PLWHA on combined 1st & 2nd line HAART | Atherosclerotic cardiovascular disease | - Prevalence of 10-year ASCVD risk was 28.7% (95% CI: 25.7–33.8%), with a significantly higher prevalence observed in the second-line combined HAART<br>- Second-line combined HAART, detectable viral load, alcohol use, and being divorced or widowed were significantly associated with 10-year ASCVD risk. |
| (Jemal et al., 2025) [33] | Adult PLWHA taking dolutegravir | Hypertension | - Prevalence of hypertension was 15.2% (95% CI: 11.9–19).<br>- Sex, duration of therapy, family history, body mass index (BMI) and fasting blood glucose level (FBG) were significantly associated with hypertension. |
| (Hirigo et al., 2024) [40] | Adult PLWHA taking dolutegravir | Hypertension High Blood Pressure (HBP) | - Prevalence of HBP: 57.9%; prehypertension: 40.5%; and hypertension: 17.3%.<br>- Initiating or switching to DTG-based HAART, being male, age > 45 years, high waist-to-height ratio, inadequate vegetable intake, low physical activity, and LDL-cholesterol was associated with higher odds of HBP. |
| (Haile et al., 2024) [27] | Adult PLWHA on HAART | Hypertension | - Prevalence of hypertension was 37.5% (95% CI: 32.8–42.5).<br>- Age groups 35–50 years, alcohol consumption, no physical exercise, family history, duration of ART, low CD4 count, and BMI ≥ 25 were associated with hypertension. |
| (Dechasa et al., 2024) [43] | Adult PLWHA on HAART | Hypertension | - Prevalence of hypertension was 23%.<br>- Rural residence, obesity, smoking and HAART regimen change were significantly associated with hypertension |
| (Befkadu et al., 2024) [41] | Adult PLWHA and HIV-negative controls | ECG abnormalities | - ECG abnormality found in 49% of HIV-infected and 19.8% of HIV-negative.<br>- The proportion of coded ST-segment abnormalities, T-wave abnormalities, longer QT interval, and sinus tachycardia was significantly higher in HIV-infected respondents.<br>- Being a smoker, being on protease inhibitors and having CD4 less than 350 cells/mm3 were significantly associated with ECG abnormalities. |

**Abbreviations:** AIDS: Acquired Immunodeficiency Syndrome; ART: Antiretroviral Treatment; BMI: Body Mass Index; CHF: Congestive Heart Failure; ECG: Electrocardiogram; FBG: Fasting Blood Sugar; HAART: Highly Active Antiretroviral Therapy; HEPI: Health Professionals Education Partnership Initiative; HIV: Human Immunodeficient Virus; PLWHA: People living with HIV/AIDS; RHD: Rheumatic Heart Disease (RHD): WHO: World Health Organization.

## Factors associated with cardiovascular disease among PLWHA

Most studies reported similar factors associated with hypertension. The most frequently reported factors were sociodemographic factors such as male gender, old age, and rural residency; behavioural factors such as high levels of alcohol consumption, cigarette smoking, and insufficient physical activity; metabolic risk factors such as higher waist circumference, waist-to-hip ratio, body mass index (BMI), blood glucose and total cholesterol; disease-related factors such as current CD4 count and higher WHO stage; treatment-related factors such as the longer duration on HAART, type of HAART and taking concomitant other drug therapy; and family history of hypertension [28–34,37,38,40,42,43]. One study also reported that elementary educational status, compared to no education, and moderate monthly income, compared to low monthly income, were significantly associated with hypertension in PLWHA [30].

Other studies have found different factors associated with various cardiovascular disorders. A study on cardiac disorders found that lower CD4 counts, not starting on ART, and lower BMI were associated with dilated cardiomyopathy, and ART status was associated with pericardial effusion [44]. In studies assessing predictors of atherosclerotic cardiovascular disease, younger age, lower systolic blood pressure, second-line combined HAART, detectable viral load, alcohol use, and lower total cholesterol were associated with atherosclerotic cardiovascular disease [22,26].

In a study of echocardiographic abnormalities, various factors were reported to be associated with specific abnormalities. Hypertension, left ventricular hypertrophy, and older age predicted diastolic dysfunction abnormality; male gender, older age, and higher fasting blood glucose level predicted ischemic heart disease; elevated blood pressure and older age predicted left ventricular hypertrophy; and hypertension predicted the presence of an enlarged left atrium. On the contrary, the severity of HIV infection did not contribute significantly to echocardiography findings [23]. Another study on the general cardiovascular effects of HIV exposure found no statistically significant differences between the HIV-exposed and non-exposed children [21]. Similarly, a study on pulmonary hypertension found no significant differences in gender, cigarette smoking, previous history of pulmonary tuberculosis treatment, chronic obstructive pulmonary disease or bronchial asthma, duration of antiretroviral therapy, or antiretroviral regimen type between PLWHA with or without pulmonary hypertension [25].

One case report also presented the case of a 19-month-old HIV-positive boy from Addis Ababa who developed an ischemic stroke following disseminated extrapulmonary tuberculosis. The study suggests extrapulmonary tuberculosis to be considered a cause for sudden focal neurologic deficits in children and adolescents with HIV infection, especially in tuberculosis-endemic countries [20] (Table 2).

## Discussion

To the best of our knowledge, this scoping review represents the first systematic synthesis of available evidence on cardiovascular diseases among PLWHA in Ethiopia. We included 26 institution-based studies, predominantly conducted in three of the nine regions, focusing largely on hypertension but also covering conditions such as atherosclerotic cardiovascular disease, ischemic stroke, diastolic dysfunction, left ventricular hypertrophy, and pericardial effusion. The majority of studies were cross-sectional, conducted after 2011, and had relatively small sample sizes. These findings underscore a growing but still limited and fragmented body of literature on cardiovascular disorders in the Ethiopian HIV population. The disproportionate focus on hypertension may reflect its relatively straightforward diagnosis using basic clinical tools such as blood pressure cuffs, compared with other cardiovascular disorders that require more advanced diagnostics like echocardiography or laboratory biomarkers.

The reported prevalence of hypertension among PLWHA in Ethiopia ranged widely from 11.0% to 41.3%, which could be attributed to variations in diagnostic thresholds, participant characteristics (e.g., HAART status), comorbidities, and institutional differences in clinical practice. This heterogeneity underscores the need for standardised diagnostic criteria and robust surveillance systems for hypertension and other cardiovascular comorbidities among PLWHA in Ethiopia. Our

findings align with other regional and global studies. For instance, two recent meta-analyses reported the global hypertension prevalence rates of 25.2% and 23.6% among PLWHA [46,47]. In East Africa, a recent meta-analysis estimated the pooled prevalence in Ethiopia at 16.13% [48]. These results suggest that Ethiopia's rates are within global estimates but also reflect underrepresentation due to the limited number and scope of existing studies. The lack of community-based studies in Ethiopia limits the generalizability of findings, as institution-based research might not reflect the true prevalence due to selection bias. Furthermore, according to the hypertension treatment guideline, in Ethiopia, hypertension is diagnosed if, on two visits on different days, systolic blood pressure is ≥ 140 mmHg and/or diastolic blood pressure is ≥ 90 mmHg on both days. However, the studies included in these reviews used various definitions of hypertension, which requires careful interpretation of the findings.

Several modifiable and non-modifiable risk factors were consistently associated with hypertension, including older age, male sex, alcohol consumption, physical inactivity, elevated body mass index (BMI), central obesity, and prolonged exposure to HAART regimens. Similar associations have been noted in studies from Burundi and other East African countries [48,49]. However, given the predominance of cross-sectional designs, causal inferences remain limited. Prospective cohort studies are urgently needed to better delineate the temporal associations between HIV-related risk factors and cardiovascular outcomes.

The high burden of hypertension and other cardiovascular complications among PLWHA calls for the integration of cardiovascular screening into routine HIV care in Ethiopia, especially for patients with modifiable risk factors. Evidence from high-income countries and emerging programs in sub-Saharan Africa suggests that integrated care models can improve outcomes and reduce costs [50,51]. Preventive strategies such as promoting physical activity, healthy diets, and blood pressure monitoring should be prioritised. In particular, patients on ART are at elevated risk of developing metabolic syndromes, including hypertension, due to both the effects of ART and ageing.

High rates of uncontrolled hypertension reported in some studies further highlight gaps in follow-up care and adherence monitoring. Strengthening chronic care delivery through regular follow-up, task-shifting, and community health worker involvement could mitigate these issues. Additionally, the Ethiopian Ministry of Health should consider updating national HIV treatment guidelines to incorporate cardiovascular screening and prevention strategies.

Despite the growing number of studies in recent years, significant research gaps remain. For instance, few studies explored conditions like ischemic stroke, cardiomyopathy, or pulmonary hypertension in depth, even though emerging evidence suggests these are increasingly relevant complications in long-term HIV care. Moreover, almost all studies were limited to urban health facilities, with an overrepresentation of regions such as Addis Ababa, Amhara, and SNNPR, leaving rural areas understudied. Future research should prioritise community-based epidemiological studies to determine the actual prevalence and burden of cardiovascular disease among PLWHA; longitudinal studies to better understand causal relationships and disease progression; and the evaluation of the effectiveness of integrated HIV-cardiovascular disease management models in Ethiopian healthcare settings.

This scoping review has strengths and limitations. The study is the first in Ethiopia to summarise evidence on cardiovascular diseases in PLWHA. We tried to look at the relationship between HIV/AIDS and any cardiovascular diseases without restriction. We extensively searched both grey and peer-reviewed literature from 6 different sources. Additionally, there were no restrictions on study design or on the time frame for article inclusion. However, our study also has limitations that should be considered while interpreting the results. Most of the included primary studies were cross-sectional and institution-based studies. Therefore, the prevalence reported in those studies might not represent the community's prevalence. Additionally, the review was restricted to English-language studies, which may have excluded relevant studies in other languages, although we do not expect Ethiopian studies to be published in languages other than English. We recommend that future studies utilise a community-based approach and employ diverse study designs, including longitudinal and interventional designs that support causal inference.

## Conclusion

This scoping review highlights a growing body of evidence on cardiovascular diseases among people living with HIV/AIDS in Ethiopia, with a predominant focus on hypertension. The findings reveal a wide range of hypertension prevalence and identify several common risk factors, including age, sex, lifestyle behaviours, metabolic indicators, and antiretroviral therapy-related variables. However, the evidence remains limited in scope, with most studies being institution-based, cross-sectional, and concentrated in a few regions of the country. Other significant cardiovascular conditions, such as atherosclerotic disease, cardiomyopathies, and stroke, have received minimal research attention.

The findings underscore the need for more comprehensive, community-based, and longitudinal studies to better understand the true burden and determinants of cardiovascular diseases among PLWHA in Ethiopia. There is also a clear call for integrating cardiovascular disease screening and preventive strategies into routine HIV care to mitigate the growing double burden of infectious and non-communicable diseases. Strengthening surveillance, diagnostic capacity, and awareness among healthcare providers will be critical for early detection and management of cardiovascular diseases in this vulnerable population.

## Supporting information

**S1 File. Search strategy.**
(DOCX)

**S2 File. PRISMA-ScR Checklist.**
(DOCX)

**S3 File. List of excluded articles.**
(DOCX)

## Acknowledgments

We thank the School of Public Health, Liverpool John Moores University, for providing the opportunity to conduct this study as a partial fulfilment of the requirements for the Master of Public Health (General MPH) for the primary author (DE).

## Author contributions

**Conceptualization:** Dawit Errega, Marios Kantaris.

**Data curation:** Dawit Errega, Kaleab Berhanu, Sisay Mulugeta Alemu.

**Formal analysis:** Dawit Errega, Andualem Deresse, Yohannes Gebreegziabhere, Sisay Mulugeta Alemu.

**Investigation:** Andualem Deresse, Kaleab Berhanu, Yohannes Gebreegziabhere, Sisay Mulugeta Alemu.

**Methodology:** Dawit Errega, Marios Kantaris, Andualem Deresse, Kaleab Berhanu, Yohannes Gebreegziabhere, Sisay Mulugeta Alemu.

**Project administration:** Dawit Errega, Kaleab Berhanu.

**Software:** Dawit Errega, Andualem Deresse, Yohannes Gebreegziabhere, Sisay Mulugeta Alemu.

**Supervision:** Marios Kantaris, Kaleab Berhanu, Yohannes Gebreegziabhere, Sisay Mulugeta Alemu.

**Validation:** Marios Kantaris, Andualem Deresse, Kaleab Berhanu, Sisay Mulugeta Alemu.

**Visualization:** Andualem Deresse, Kaleab Berhanu, Yohannes Gebreegziabhere, Sisay Mulugeta Alemu.

**Writing – original draft:** Dawit Errega, Sisay Mulugeta Alemu.

**Writing – review & editing:** Dawit Errega, Marios Kantaris, Andualem Deresse, Kaleab Berhanu, Yohannes Gebreegziabhere, Sisay Mulugeta Alemu.

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
