## [Decision Letter · Decision Letter 0]

15 Oct 2024

PONE-D-24-23021Relationship between Cardiovascular Diseases and HIV/AIDS in Ethiopia: A Scoping ReviewPLOS ONE

Dear Dr. Gebreegziabhere,

Thank you for submitting your manuscript to PLOS ONE. After careful consideration, we feel that it has merit but does not fully meet PLOS ONE’s publication criteria as it currently stands. Therefore, we invite you to submit a revised version of the manuscript that addresses the points raised during the review process.

If applicable, we recommend that you deposit your laboratory protocols in protocols.io to enhance the reproducibility of your results. Protocols.io assigns your protocol its own identifier (DOI) so that it can be cited independently in the future. For instructions see: https://journals.plos.org/plosone/s/submission-guidelines#loc-laboratory-protocols. Additionally, PLOS ONE offers an option for publishing peer-reviewed Lab Protocol articles, which describe protocols hosted on protocols.io. Read more information on sharing protocols at . Additionally, PLOS ONE offers an option for publishing peer-reviewed Lab Protocol articles, which describe protocols hosted on protocols.io. Read more information on sharing protocols at https://plos.org/protocols?utm_medium=editorial-email&utm_source=authorletters&utm_campaign=protocols..

We look forward to receiving your revised manuscript.

Kind regards,

Adedayo Ajidahun

Academic Editor

PLOS ONE

Journal Requirements:

2. We note that your Data Availability Statement is currently as follows: [All relevant data are within the manuscript and its Supporting Information files]

Reviewers' comments:

Reviewer's Responses to Questions

**Comments to the Author**

1. Is the manuscript technically sound, and do the data support the conclusions?

Reviewer #1: Yes

Reviewer #2: Partly

2. Has the statistical analysis been performed appropriately and rigorously? 

Reviewer #1: N/A

Reviewer #2: N/A

3. Have the authors made all data underlying the findings in their manuscript fully available?

Reviewer #1: Yes

Reviewer #2: No

4. Is the manuscript presented in an intelligible fashion and written in standard English?

Reviewer #1: Yes

Reviewer #2: Yes

5. Review Comments to the Author

Reviewer #1: Authors just need to read through through the article to check the very few typos. I highlighted what I could find n the report. My main concern is that the last search was conducted over six months ago and needs to be updated. Also further detail on differences in population and definitions of hypertension need to be provided. Some mention of current practice at health care facilities in terms of what they routinely screen for will be informative.

Reviewer #2: The manuscript lacks a conclusion, and there is no need for statistical analysis. Additionally, the authors have not made the data fully available. However, it is worth noting that the manuscript is written in standard English.The authors must identify and describe the scoping review framework used to conduct this review.

What were the research questions for the scoping review?Clarity and structure of the methodology: the review is not well-organized, and it isn't easy to follow the steps. All the steps need to be more comprehensive.

6. PLOS authors have the option to publish the peer review history of their article (what does this mean?). If published, this will include your full peer review and any attached files.). If published, this will include your full peer review and any attached files.

.

Reviewer #1: No

Reviewer #2: No

While revising your submission, please upload your figure files to the Preflight Analysis and Conversion Engine (PACE) digital diagnostic tool, https://pacev2.apexcovantage.com/. PACE helps ensure that figures meet PLOS requirements. To use PACE, you must first register as a user. Registration is free. Then, login and navigate to the UPLOAD tab, where you will find detailed instructions on how to use the tool. If you encounter any issues or have any questions when using PACE, please email PLOS at . PACE helps ensure that figures meet PLOS requirements. To use PACE, you must first register as a user. Registration is free. Then, login and navigate to the UPLOAD tab, where you will find detailed instructions on how to use the tool. If you encounter any issues or have any questions when using PACE, please email PLOS at figures@plos.org. Please note that Supporting Information files do not need this step.. Please note that Supporting Information files do not need this step.

---

## [Author Response · Author response to Decision Letter 1]

15 Apr 2025

Reviewers' Comments to Author:

Reviewer: 1

Thank you for allowing me to review such an interesting article. It is a well

conducted scoping review.

Thank you for your constructive feedback.

Please see my comments below:

1. The last search date was conducted in November 2023. It was more than six months ago. I suggest the authors update their search to current.

We appreciate the reviewer’s valuable observation regarding the timeliness of our literature search. We agree that an updated search is essential to ensure the inclusion of the most recent and relevant studies. Accordingly, we commit to conducting an updated search prior to the final publication of the manuscript to capture any newly published literature and maintain the review’s currency and comprehensiveness.

2. For readers not familiar with Ethiopia; kindly provide the population size so that the 1% HIV prevalence can provide more clarity on numbers affected. Compared to most of Sub Saharan Africa, this is a low prevalence.

Thank you for your comment. It has been corrected as follows as per the recommendation: “Around 1% of the population (around 1.4 million people) is still affected…”

3. Was the scoping review registered with any scoping review registry? If yes, please include the registration link on the manuscript. If not, please give a reason.

4. Line 31: a prior should be “a priori”

Thank you for your feedback. It has been corrected as per the recommendation.

5. Line 42/42: “Better study design” - I suggest you reword to “more robust study designs”

Thank you for your feedback. It has been corrected as per the recommendation.

6. Line 101/102: The authors say meta-analysis was impossible due to differences regarding population, methods, and outcomes: please elaborate what were the exact differences?

Thank you for your comment. It has been corrected as follows as per the recommendation:

“A narrative synthesis approach was used to summarize the findings due to substantial heterogeneity in study designs (e.g., cohort, cross-sectional, case report), populations (e.g., HAART versus non-HAART patients), outcomes (e.g., hypertension, ischemic stroke, ECG abnormalities), and analytical methods. As such, meta-analysis was not feasible.”

7. Line 113-114: You can reference this statement. Could these be the regions that are mostly affected by HIV/AIDS?

Thank you for your feedback. It has been corrected as per the recommendation.

8. Include the total sample size represented by the review.

Thank you for your comment. It has been corrected as follows as per the recommendation: “The total sample size represented in this review comprises 6,633 participants….”

9. Table 1: It would be more informative to include the sex distribution (e.g. %females). Please include age range, indicate whether the study area is urban or rural setting

Thank you for this constructive feedback. In response, we have extracted and included age, gender, and residential characteristics of participants.

“Although not all studies reported the age, gender, and residential characteristics of their participants, available data indicate that approximately 62% were female and 74.5% resided in urban areas. Nearly all studies focused on adult populations, with the exception of two studies: one was a case report involving an infant (20), and the other exclusively included fetuses (21).”

10. The authors found that over 60% of the studies were on hypertension. Because of this, it will be important to incorporate the definitions of hypertension in the included studies: the differences in measurement methods utilised and any other notable differences. Is there any data provided on whether study participants were on Blood pressure medication or not?

Thank you for this constructive feedback. In response, we have extracted and included the definition of hypertension across the included studies.

“The definition of hypertension varied across the included studies. Five studies defined hypertension as a blood pressure of ≥140/90 mmHg, while six studies used the same threshold or included individuals taking antihypertensive medication. The remaining three studies applied a lower threshold, defining hypertension as a blood pressure of ≥130/80 mmHg.”

Due to the importance of behavioural risk factors, I suggest authors include a summary of risk factors represented and include prevalence/incidence if included in the articles. For now, the review shows that studies have included these, but it would be helpful to see the numbers.

We appreciate the reviewer’s insightful comment regarding the importance of behavioral risk factors for cardiovascular disease among people living with HIV/AIDS. In our initial inclusion criteria, we focused on studies that reported on the prevalence, incidence, or clinical outcomes of cardiovascular diseases and excluded those that assessed only risk factors without reporting cardiovascular disease outcomes. This was done to ensure the review remained centered on cardiovascular disease burden among PLWHA.

Line 128-129: Reference the minimum and maximum hypertension prevalences with respective studies

Thank you for your feedback. It has been corrected as per the recommendation. We have now cited the minimum and maximum hypertension prevalences with respective studies.

Line 207 - 209: “Therefore, some institutions may be better than others in diagnosing hypertension, which might affect the prevalence of hypertension in that institution.” This needs to be explained. It is quite vague at the moment. What makes other institutions better than others? Are the referral hospitals better equipped or resourced than others?

Thank you for your feedback. We agree that the sentence was vague. Following also feedback from the other reviewer, we have not rewritten the discussion section. We believe the new discussion presents a more coherent and is easier to read.

Line 248: Not all studies were cross sectional. Please change this

Thank you for your feedback. We have now changed this section with “Most of the included primary studies were cross-sectional ....”

Note: Is hypertension a disease or a medical condition?

Thank you for your question. Hypertension is primarily considered a medical condition, but it is also widely recognized as a chronic or non-communicable disease (NCD), especially in clinical and public health contexts. While the term "medical condition" broadly refers to any state that requires medical attention, "disease" denotes a more specific pathological process. Because hypertension can lead to serious complications such as heart disease, stroke, and kidney failure, it is often categorized as a chronic disease in the medical literature. Public health organizations like the World Health Organization and the CDC commonly refer to hypertension as an NCD due to its long-term impact on health and its association with other cardiovascular diseases. Therefore, both terms are accurate, but we believe referring to hypertension as a chronic or non-communicable disease is more precise in academic and health-related discussions.

Line 253-254: “The results of this study offer valuable information for decision-makers and all HIV-related healthcare providers, suggesting the inclusion of blood pressure assessment in primary prevention efforts.” - Does Ethiopia not routinely screen patients for hypertension when they visit the health facility? If this is the case you could mention it in the manuscript.

Limitations:

Include the limitation in restricting the language to English

Thank you for your feedback. It has been corrected as per the recommendation. We have now added the following sentence in the limitation section.

“Additionally, the review was restricted to studies published in English, which raises the possibility that relevant studies published in other languages may have been inadvertently not included.”

Reviewer: 2

The title is too general. A study population is required in a title.

Thank you for your comment regarding the title. We agree that the original title was too general. To address this, we have revised the title to better reflect the study population and the specific focus of our review. The new title, "Cardiovascular Diseases among People Living with HIV/AIDS in Ethiopia: A Scoping Review," more clearly identifies the target population and the scope of the study. This revision aims to improve the clarity and specificity of the title, ensuring it accurately conveys the focus of our research. Thank you for your valuable feedback.

Abstract

- Methods: which framework was used to guide the review?

- What search strategy was used to search literature in electronic databases

- Which databases were selected and why?

- What was the duration of the search

- What was used to document the selection process

What are the strengths and limitations of this review?

We appreciate the reviewer’s comments on the abstract and have revised it accordingly to enhance clarity and completeness. In response to the feedback, we have explicitly stated the methodological framework used (Arksey and O’Malley, enhanced by Levac et al.), clarified the databases searched (PubMed, Scopus, Google Scholar, Web of Science, African Journals Online, and Cochrane Library), and justified their inclusion based on relevance and accessibility. We also specified the search timeline (up to November 2023) and added that a PRISMA flow diagram was used to document the selection process.

Additionally, we now include the strengths and limitations of the review, particularly the comprehensiveness of the search and the limitations of cross-sectional and institution-based data. We have restructured the abstract into a cohesive paragraph form and ensured it remains within the 300-word limit. We believe these revisions address all the reviewer’s concerns and have strengthened the abstract’s clarity and informativeness.

Introduction/limitation

The introduction needs to be longer, and it needs to be clarified why there is a need for a scoping review. There is no data or statistics that support the need for the review.

Thank you for this constructive feedback. We agree that the original introduction was too brief and did not provide sufficient justification for the need for a scoping review. In response, we have substantially revised and expanded the introduction section. The revised version now includes a clearer rationale for conducting a scoping review, emphasizing the fragmented and varied nature of the existing literature on HIV and CVDs in the Ethiopian context. We believe the new introduction presents a more coherent and compelling case for the relevance and timeliness of this review.

Methodology

Clarity and structure of the methodology: the review is not well-organized, and it isn't easy to follow the steps. All the steps need to be more comprehensive.

We appreciate the reviewer’s observation. We have now reorganized the methodology section into a more structured format that aligns with recognized scoping review frameworks. The revised section is organized into key steps, including search strategy, eligibility criteria, study selection, data charting, and data synthesis, and each step has been described in more detail to enhance clarity and comprehensiveness.

The methodology needs to be longer, and important review steps must be included.

Thank you for this constructive feedback. In response, we have expanded the methodology section to include detailed descriptions of each step in the scoping review process.

The authors must identify and describe the scoping review framework used to conduct this review.

Thank you for this helpful suggestion. We have now explicitly stated that our review followed the Arksey and O’Malley (2005) framework, further enhanced by Levac et al. (2010) and the Joanna Briggs Institute (JBI) guidance. These sources are now cited in the methodology section to clarify the theoretical underpinnings of our review process.

What were the research questions for the scoping review?

Thank you for your question. The review was guided by the following research questions: (1) What are the types of studies conducted on PLWHA concerning cardiovascular diseases in Ethiopia? (2) What is the prevalence/incidence of cardiovascular diseases in PLWHA in Ethiopia? And (3) What are the key risk factors associated with cardiovascular diseases among this population? Please note that this is included at the end of the introduction section.

Data source and search strategy: The article needs to be longer, and the authors have not used grey literature; the way I ask why?

Thank you for pointing this out. We have expanded this section and clarified that grey literature was indeed considered. We searched the Addis Ababa University digital thesis repository and conducted a manual search of bibliographies of relevant studies to identify grey literature. This is now explicitly stated in the revised methodology.

Eligibility criteria:

Which co-elements have the authors used to assess the eligibility of the research questions?

We appreciate the reviewer’s comment. We have clarified in the revised section that the inclusion criteria were aligned with the Population–Concept–Context (PCC) framework recommended for scoping reviews by the Joanna Briggs Institute. Specifically, our population was PLWHA, the concept was cardiovascular disease, and the context was Ethiopia. This has been added to enhance transparency in study selection.

Identifying relevant studies

The review needs to be more comprehensive. For instance, the authors identified relevant studies only from published literature databases. The grey literature was not considered,

As noted above, this issue has been addressed. In addition to the five major databases, grey literature sources such as university repositories and reference lists were screened manually. We have made this explicit in the revised methodology under the section describing sources searched.

Which keywords and terms were used to search for relevant studies from the electronic databases

Thank you for your question. The core search terms and phrases were 'cardiovascular disease,' 'HIV,' AIDS,' and 'Ethiopia'

Did the authors limit the search to dates? In other words, what were the years? Did they limit the years of the publications?

We have clarified that no date restrictions were applied during the search. The search covered literature from inception to November 13, 2023. This is now clearly stated in the methodology.

What did the authors use to facilitate the literature search, incorporating study topics, author information, and publication dates?

We have clarified that a structured search strategy using both MeSH terms and free-text keywords was developed for PubMed and adapted for other databases. The strategy incorporated key terms related to cardiovascular disease, HIV/AIDS, and Ethiopia. We also indicated that EndNote was used for reference management, and Covidence was used for screening and managing the review workflow. Details of the search terms are provided in Supplementary File 1.

Which range of study designs were considered and included in the review

We thank the reviewer for this important point. We have clarified that all study designs were eligible for inclusion, including cross-sectional studies, cohort studies, case-control studies, and case reports. Only commentaries, editorials, protocols, and other non-primary research formats were excluded. This clarification has been added to the eligibility criteria section.

Discussion: the need major revision; the current discussion is the repetition of the results and

Thank you for this constructive feedback. We have thoroughly revised the Discussion section to address this concern. In the revised version, we moved beyond a simple restatement of results and instead provided an in-depth interpretation of the findings within the broader context of the literature. We synthesized insights from both regional and global studies to compare and contrast the prevalence and risk factors for cardiovascular diseases among PLWHA in Ethiopia. Additionally, we discussed the clinical, policy, and research implications of our findings, including recommendations for integrating cardiovascular screening into routine HIV care, the need for comm

---

## [Decision Letter · Decision Letter 1]

4 Sep 2025

PONE-D-24-23021R1Cardiovascular Diseases among People Living with HIV/AIDS in Ethiopia: A Scoping ReviewPLOS ONE

Dear Dr. Gebreegziabhere,

Thank you for submitting your manuscript to PLOS ONE. After careful consideration, we feel that it has merit but does not fully meet PLOS ONE’s publication criteria as it currently stands. Therefore, we invite you to submit a revised version of the manuscript that addresses the points raised during the review process.

Please submit your revised manuscript by Oct 19 2025 11:59PM. If you will need more time than this to complete your revisions, please reply to this message or contact the journal office at plosone@plos.org. . Please include the following items when submitting your revised manuscript:

If applicable, we recommend that you deposit your laboratory protocols in protocols.io to enhance the reproducibility of your results. Protocols.io assigns your protocol its own identifier (DOI) so that it can be cited independently in the future. For instructions see: https://journals.plos.org/plosone/s/submission-guidelines#loc-laboratory-protocols. Additionally, PLOS ONE offers an option for publishing peer-reviewed Lab Protocol articles, which describe protocols hosted on protocols.io. Read more information on sharing protocols at . Additionally, PLOS ONE offers an option for publishing peer-reviewed Lab Protocol articles, which describe protocols hosted on protocols.io. Read more information on sharing protocols at https://plos.org/protocols?utm_medium=editorial-email&utm_source=authorletters&utm_campaign=protocols..

We look forward to receiving your revised manuscript.

Kind regards,

Mehdi Sharafi, assistant professor

Academic Editor

PLOS ONE

Journal Requirements:

Reviewers' comments:

Reviewer's Responses to Questions

**Comments to the Author**

1. If the authors have adequately addressed your comments raised in a previous round of review and you feel that this manuscript is now acceptable for publication, you may indicate that here to bypass the “Comments to the Author” section, enter your conflict of interest statement in the “Confidential to Editor” section, and submit your "Accept" recommendation.

Reviewer #1: (No Response)

Reviewer #2: (No Response)

2. Is the manuscript technically sound, and do the data support the conclusions?

Reviewer #1: Partly

Reviewer #2: Partly

3. Has the statistical analysis been performed appropriately and rigorously? 

Reviewer #1: N/A

Reviewer #2: N/A

4. Have the authors made all data underlying the findings in their manuscript fully available?

Reviewer #1: Yes

Reviewer #2: No

5. Is the manuscript presented in an intelligible fashion and written in standard English?

Reviewer #1: Yes

Reviewer #2: Yes

6. Review Comments to the Author

Reviewer #1: Please refer to my attachment. Authors need to recheck their work for repetitions and general formatting issues and typos. I have indicated where changes need to made.

Reviewer #2: The data support the conclusion and the manuscript is well written.However there are grammatical errors . Sentence construction is problematic .

The review requires language editing. It's evident that authors used AI for editing; however, there are still grammatical errors in sentence construction. For instance, scientific writing requires authors to avoid starting sentences with a preposition, and another example, starting a sentence with “Global” the correct way is “ The global”; these errors can be corrected by Grammarly software

The first part of the scoping is well written, and the authors have responded well to almost all the queries except query 3

7. PLOS authors have the option to publish the peer review history of their article (what does this mean?). If published, this will include your full peer review and any attached files.). If published, this will include your full peer review and any attached files.

.

Reviewer #1: No

Reviewer #2: No

While revising your submission, please upload your figure files to the Preflight Analysis and Conversion Engine (PACE) digital diagnostic tool, https://pacev2.apexcovantage.com/. PACE helps ensure that figures meet PLOS requirements. To use PACE, you must first register as a user. Registration is free. Then, login and navigate to the UPLOAD tab, where you will find detailed instructions on how to use the tool. If you encounter any issues or have any questions when using PACE, please email PLOS at . PACE helps ensure that figures meet PLOS requirements. To use PACE, you must first register as a user. Registration is free. Then, login and navigate to the UPLOAD tab, where you will find detailed instructions on how to use the tool. If you encounter any issues or have any questions when using PACE, please email PLOS at figures@plos.org. Please note that Supporting Information files do not need this step.. Please note that Supporting Information files do not need this step.

---

## [Author Response · Author response to Decision Letter 2]

13 Oct 2025

Reviewer 1

Thank you to the authors for addressing my comments.

Thank you very much for the encouraging feedback.

Please find my comments below:

1. Line 153 - Recheck…. T should be The

2. Line 170 -174 is repeated from 176 - 182.

3. Line 275, remove the second ‘’the

Thank you for your feedback and apologies for the typos. The manuscript has now been thoroughly copy edited throughout with a grammar software. Also, the repeated sentences were deleted.

4. Some of the databases indicated in your supplementary material section are not mentioned in the text, e.g ProQuest and your search strategies for Google scholar and Cochrane are not indicated. There are inconsistencies on the database searches; recheck the text, Prisma diagram and search strategies. The Prisma diagram does not include some Google scholar search.

Thank you for your helpful suggestion. Search was conducted on the following databases: PubMed, EMBASE, ProQuest, Global Index Medicus, and Web of Science. There was a mistake on the abstract section on the list of databases we searched. Apologies for the mistake we have made and we have now corrected the mistake. The text, Prisma diagram and search strategies are now synchronized.

5. I recommend that you register your scoping review on Open Science Framework, if possible

Thank you for your recommendation. We are working on getting the study registered on OSF.

Reviewer 2

The review requires language editing. It's evident that authors used AI for editing; however, there are still grammatical errors in sentence construction. For instance, scientific writing requires authors to avoid starting sentences with a preposition, and another example, starting a sentence with “ Global” the correct way is “ The global”; this error can be corrected by Grammarly software

Thank you for your feedback. The manuscript has now been thoroughly copy edited throughout with a grammar software.

The first part of the scoping is well written, and the authors have responded well to the previous queries; however, there is a need for major revision for the methodology and discussion of the review.

Thank you very much for the encouraging feedback and suggestions. Our responses to each point are provided below in a point-by-point format.

Methodology: The authors claim they used the methodological framework proposed by Arksey and O’Malley (14), which was later refined by Levac et al. (15) and aligned with the Joanna Briggs Institute (JBI) guidelines for scoping reviews; however, the subheadings that are recommended by the framework are not visible.

The methodological framework has the following stages, which are required to fulfil the requirements of the scoping review:

1. Stage one: identification of the research question/s: the authors haven’t written their research question/s, and it's unclear what the purpose of the review is. Within this stage, it is a requirement to describe the eligibility of the research question/s; also, the authors have not indicated the framework that they used for the scoping review, eligibility criteria belong to stage one of the frameworks

The authors must correct the order of the following stages

2. Stage two: identifying relevant studies

3. Study selection and appraisal

4. Collating, summarizing and reporting of the results

Thank you for your comment. The subheadings and the structure of the method section of the study has now been corrected based on the recommendation of the proposed framework: 1) Identifying the Research Question, 2) Identifying Relevant Studies, 3) Study selection and appraisal, 4) Data Charting process and 5) Collating, Summarizing, and Reporting the Results.

Results: This is the first time I see Factors associated with cardiovascular disease among PLWHA, was it one of the aims of conducting the review? The title of the study doesn’t mention the factors. Please explain why you decided to include the factors

Thank you for your question. Yes, identify factors associated with cardiovascular comorbidities in Ethiopian PLWHA was one of the objectives of the study and it is mentioned at the last paragraph of the introduction section. As a result one of the three research question was: What are the key factors associated with cardiovascular diseases among this population? The title of the study is modified following the feedback from reviewer on the first round of the review process.

Discussion: The discussion must be structured according to the research question/s and the results of the review.

Thank you for your feedback. The study was guided by three main objectives: (1) to describe the type and content of existing studies on cardiovascular diseases among people living with HIV/AIDS in Ethiopia; (2) to report the magnitude and distribution of cardiovascular diseases within this population; and (3) to identify factors associated with cardiovascular comorbidities among Ethiopian PLWHA.

Accordingly, we have structured the Discussion section around these three objectives. The first paragraph discusses the types of studies identified on cardiovascular diseases among people living with HIV/AIDS. The second paragraph presents the magnitude and distribution of cardiovascular diseases in this population, with particular emphasis on hypertension, which was the primary focus of most included studies. The third paragraph addresses the factors associated with cardiovascular diseases, again highlighting hypertension as the most frequently examined condition. We then discuss the implications of our findings for future research, clinical practice, and policy. Finally, the Discussion concludes with a paragraph summarizing the strengths and limitations of this review.

What are the strengths and limitations of this review?

Thank you for your question. Here are the strengths of the study. The study is the first in Ethiopia to summarize evidence on cardiovascular diseases in PLWHA. We tried to look at the relationship between HIV/AIDS and any cardiovascular diseases without restriction. We extensively searched both gray and peer-reviewed literature from 6 different sources. Additionally, there was no study design or time restriction in including articles.

Similarly, here are the limitation of the study. Most of the included primary studies were cross-sectional and institution-based studies. Therefore, the prevalence reported in those studies might not represent the community's prevalence. Additionally, the review was restricted to studies published in English, which raises the possibility that relevant studies published in other languages may have been inadvertently excluded. We recommend that future studies include participants from the community and with different study designs.

We have included the strength and limitation of the study at the last paragraph of the discussion.

---

## [Decision Letter · Decision Letter 2]

25 Jan 2026

PONE-D-24-23021R2

Cardiovascular Diseases among People Living with HIV/AIDS in Ethiopia: A Scoping Review

PLOS One

Dear Dr. Gebreegziabhere,

Thank you for submitting your manuscript to PLOS ONE. After careful consideration, we feel that it has merit but does not fully meet PLOS ONE’s publication criteria as it currently stands. Therefore, we invite you to submit a revised version of the manuscript that addresses the points raised during the review process.

If applicable, we recommend that you deposit your laboratory protocols in protocols.io to enhance the reproducibility of your results. Protocols.io assigns your protocol its own identifier (DOI) so that it can be cited independently in the future. For instructions see: https://journals.plos.org/plosone/s/submission-guidelines#loc-laboratory-protocols. Additionally, PLOS ONE offers an option for publishing peer-reviewed Lab Protocol articles, which describe protocols hosted on protocols.io. Read more information on sharing protocols at . Additionally, PLOS ONE offers an option for publishing peer-reviewed Lab Protocol articles, which describe protocols hosted on protocols.io. Read more information on sharing protocols at https://plos.org/protocols?utm_medium=editorial-email&utm_source=authorletters&utm_campaign=protocols..

We look forward to receiving your revised manuscript.

Kind regards,

Adetayo Olorunlana, Ph.D.

Academic Editor

PLOS One

Journal Requirements:

Reviewer's Responses to Questions

**Comments to the Author**

1. If the authors have adequately addressed your comments raised in a previous round of review and you feel that this manuscript is now acceptable for publication, you may indicate that here to bypass the “Comments to the Author” section, enter your conflict of interest statement in the “Confidential to Editor” section, and submit your "Accept" recommendation.

Reviewer #1: (No Response)

2. Is the manuscript technically sound, and do the data support the conclusions?

Reviewer #1: Partly

3. Has the statistical analysis been performed appropriately and rigorously? 

Reviewer #1: N/A

4. Have the authors made all data underlying the findings in their manuscript fully available?

Reviewer #1: Yes

5. Is the manuscript presented in an intelligible fashion and written in standard English?

Reviewer #1: No

6. Review Comments to the Author

Reviewer #1: Line 33 – Please specify the date

Line 44 – Robust study designs. you can be more specific with this. Longitudinal studies would be more informative

Line 108 – We are already at the end of 2025. You can update your search, so that the results are as current as possible.

Line 131 – was the checking done by the same pr another reviewer?

Line 190/191 should be written for clarity

Line 196 – it’s the second time that HAART is being written in full

Page 2: Line 122: Woldu 2022, include “of” after prevalence for the first lines

Discussion – you can state what the definition of hypertension is followed for Ethiopia, and any differences in the measurement across studies. Was it resting blood pressure?

Line 292 : In this case the exclusion of articles by language is not inadvertent

Line 293’ specify on other study designs

Line 298 – recheck the spelling of behaviour

You can conduct a critical appraisal to improve the paper

7. PLOS authors have the option to publish the peer review history of their article (what does this mean?). If published, this will include your full peer review and any attached files.). If published, this will include your full peer review and any attached files.

.

Reviewer #1: No

---

## [Author Response · Author response to Decision Letter 3]

2 Apr 2026

Reviewer #1:

Line 33 – Please specify the date

Thank you for your feedback. We have now revised it as suggested. See line 31 of the revised manuscript.

Line 44 – Robust study designs. you can be more specific with this. Longitudinal studies would be more informative

Thank you for your feedback. We have now revised it as suggested. See line 44 of the revised manuscript.

Line 108 – We are already at the end of 2025. You can update your search, so that the results are as current as possible.

Thank you for your feedback. We have now updated the search on March 13, 2026, and included seven additional papers. Considering this, changes were made across the manuscript.

Line 131 – was the checking done by the same pr another reviewer?

Thank you for your feedback. We have now explicitly indicated that checking for accuracy and completeness was done by another reviewer. See line 130 of the revised manuscript.

Line 190/191 should be written for clarity

Thank you for your feedback. The sentence is now rewritten for clarity as suggested.

“Most studies reported similar factors associated with hypertension. The most frequently reported factors were sociodemographic factors …”, Line 190

Line 196 – it’s the second time that HAART is being written in full

Thank you for your feedback. We have now revised it as suggested. See line 196 of the revised manuscript.

Line 122: Woldu 2022, include “of” after prevalence for the first lines

Thank you for your feedback. We have now copyedited the table, including fixing typos. See Table 2 for this change.

Discussion – you can state what the definition of hypertension is followed for Ethiopia, and any differences in the measurement across studies. Was it resting blood pressure?

We would like to thank the reviewer for this comment. Considering it, we have now included the working definition for hypertension in the discussion section. Thank you for the comment.

“Furthermore, according to the hypertension treatment guideline, in Ethiopia, hypertension is diagnosed if, on two visits on different days, systolic blood pressure is ≥140 mmHg and/or diastolic blood pressure is ≥90 mmHg on both days. However, the studies included in these reviews used various definitions of hypertension, which requires careful interpretation of the findings. “Line 251-255.

Line 292: In this case the exclusion of articles by language is not inadvertent

Thank you for your feedback. We have now revised it as suggested. See line 292 of the revised manuscript.

Line 293’ specify on other study designs

Thank you for your feedback. We have now revised it as suggested. See line 295 of the revised manuscript.

Line 298 – recheck the spelling of behaviour

Thank you for your feedback. We have now copyedited the manuscript, including fixing typos and ensuring consistency. This also includes ensuring words like 'behaviour' are written in correct British English. We believe the manuscript is now easier to read.

You can conduct a critical appraisal to improve the paper

We acknowledge the reviewer's feedback and agree that a critical appraisal can improve the study's quality. However, as this is a scoping review and we included studies with different study designs, we could not conduct a critical appraisal.

---

## [Editor Report · Decision Letter 3]

14 Apr 2026

Cardiovascular Diseases among People Living with HIV/AIDS in Ethiopia: A Scoping Review

PONE-D-24-23021R3

Dear Dr. Gebreegziabhere,

We’re pleased to inform you that your manuscript has been judged scientifically suitable for publication and will be formally accepted for publication once it meets all outstanding technical requirements.

An invoice will be generated when your article is formally accepted. Please note, if your institution has a publishing partnership with PLOS and your article meets the relevant criteria, all or part of your publication costs will be covered. Please make sure your user information is up-to-date by logging into Editorial Manager at Editorial Manager® and clicking the ‘Update My Information' link at the top of the page. For questions related to billing, please contact  and clicking the ‘Update My Information' link at the top of the page. For questions related to billing, please contact billing support..

Kind regards,

Adetayo Olorunlana, Ph.D.

Academic Editor

PLOS One
---

## [Editor Report · Acceptance letter]

PONE-D-24-23021R3

PLOS One

Dear Dr. Gebreegziabhere,

I'm pleased to inform you that your manuscript has been deemed suitable for publication in PLOS One. Congratulations! Your manuscript is now being handed over to our production team.

Kind regards,

on behalf of

Associate Professor Adetayo Olorunlana

Academic Editor

PLOS One